# Interrogating Plant-Microbe Interactions with Chemical Tools: Click Chemistry Reagents for Metabolic Labeling and Activity-Based Probes

**DOI:** 10.3390/molecules26010243

**Published:** 2021-01-05

**Authors:** Vivian S. Lin

**Affiliations:** Biological Sciences Division, Pacific Northwest National Laboratory, Richland, WA 99354, USA; vivian.lin@pnnl.gov

**Keywords:** chemical biology, plant–microbe interactions, metabolic labeling, click chemistry, biosensors, activity-based probes

## Abstract

Continued expansion of the chemical biology toolbox presents many new and diverse opportunities to interrogate the fundamental molecular mechanisms driving complex plant–microbe interactions. This review will examine metabolic labeling with click chemistry reagents and activity-based probes for investigating the impacts of plant-associated microbes on plant growth, metabolism, and immune responses. While the majority of the studies reviewed here used chemical biology approaches to examine the effects of pathogens on plants, chemical biology will also be invaluable in future efforts to investigate mutualistic associations between beneficial microbes and their plant hosts.

## 1. Introduction

A diverse array of chemical biology tools and approaches has been developed for investigating biological questions in a wide variety of systems, spanning single-cell organisms to plant and animal models, as well as humans. Increasingly, researchers are delving into systems of higher and higher complexity, including microbiomes containing hundreds to thousands of different species, many of which have not yet been identified. Given this ongoing exploration, the flexible and versatile chemical biology toolbox offers unique opportunities to elucidate biological processes even in poorly characterized systems, such as gut and soil microbiomes [1,2,3].

The critical role that the plant microbiome plays in plant health is well-recognized, but we frequently lack a fundamental understanding of plant–microbe interactions at the molecular level. Microbes can be found in all regions of the plant, including the roots, stems, leaves, and seeds. Microbes may colonize tissue surfaces (epiphytes), but may also be present within plant tissues (endophytes) [4,5]. The spatial organization of these microbes may be very heterogeneous, even within the same tissue type; the biochemical pathways and other factors that help shape the root and leaf microbiomes [4,5] are poorly understood and require further exploration. Bacteria and fungi may benefit or compete with their plant hosts and other microbes depending on various environmental factors [6,7,8], making it challenging to gain a predictive understanding of how microbial communities impact plant health under different abiotic and biotic stressors [9].

At the heart of plant–microbe interactions are chemical and biochemical signals, ranging from nutrient sensing and acquisition, to growth and development, to defense and immune responses [10,11]. As a relatively young field, chemical biology encompasses a broad range of tools and approaches focused on using chemistry to gain a deeper understanding of biological processes; advances in the chemical biology toolbox, therefore, present new avenues for interrogating the diverse signaling pathways and chemical dynamics between plants and microbes. Combined with genetic, biochemical, and molecular biology methods, chemical biology will continue to play an important role in uncovering the fundamental molecular mechanisms driving these complex interkingdom interactions.

## 2. Chemical Biology Toolbox

This review will survey click chemistry reagents for metabolic labeling and activity-based probes (ABPs) that have been used for interrogating plant-microbe interactions (Figure 1). The examples herein represent only a small selection of the currently available tools and strategies in chemical biology that can be applied to plant and/or microbial systems [12]. Specifically, this review focuses on recent work that aims to interrogate interkingdom interactions between plants and microbes, with relatively brief mention of studies on plant or microbial systems in isolation. The use of metabolic labeling and additional tools and approaches for studying translation in plants have also been reviewed by Mazzoni-Putnam and Stepanova [13], and ABPs applied to plant systems have been reviewed previously in excellent detail by Overkleeft, van der Hoorn, and colleagues [14,15].

### 2.1. Bioorthogonal Click Chemistry Reagents for Metabolic Labeling

Chemical biology has relied upon bioorthogonal reagents and reactions to investigate a diverse range of systems, with ongoing exploration of new strategies and chemistries that are compatible with the biological milieu [16,17,18]. Among the most frequently used bioorthogonal tags, azides and alkynes are relatively small and generally inert functional groups that serve as synthetic handles for downstream covalent attachment of reporter groups using copper-catalyzed click chemistry or copper-free strain-assisted click chemistry [19]. This strategy is particularly valuable when introducing the reporter tag may interfere with metabolic uptake and/or label incorporation due to reporter size or chemical properties, such as molecular charge or water solubility. Reporters vary depending on the type of analysis that will be performed on the sample. Most commonly, fluorophores are attached for fluorescence detection and imaging, while conjugation with biotin allows for streptavidin affinity purification, which enriches the sample for labeled species, thereby improving detection of low abundance analytes. Click reactions can proceed in aqueous solution under mild conditions, making this chemistry highly useful for applications in complex biological mixtures [17].

Metabolic labeling of plants and microbes using bioorthogonal click chemistry reagents has led to the development of a broad range of tools, including synthetic analogs of monolignols [20,21,22,23], glycans [24,25,26], amino acids [27,28,29], and fatty acids [30,31,32] for investigating the production or modification of various biomolecular species. These bioorthogonal reagents have been designed to mimic the structure of native compounds and contain an alkyne or azide functionality for downstream attachment of reporter tags using click chemistry (Figure 2). While both alkynyl and azido reagents are generally compatible with microbial systems, alkynes may be better tolerated in plants than azido reagents [30]. Reagent stability in the experimental system of interest is an important consideration to avoid loss or modification of labeled targets. Incorporation of the labeling reagents into live cells, tissues, or whole organisms can be tracked spatially, using fluorescence imaging, as well as temporally, use pulse-chase methods.

Clickable sugars for glycan labeling in plants have been used for studying glycan metabolism and transport in specific tissues, as well as tissue structure and development [24,25]. Anderson and coworkers demonstrated the application of a commercially available alkynylated fucose analog (FucAl) for probing polysaccharide delivery and re-organization in plant tissues [25]. After metabolic labeling of *Arabidopsis* seedlings followed by click chemistry to attach fluorophores for fluorescence imaging, time-dependent distribution of the labeling in root cells was observed. Anderson and coworkers determined that a significant amount of the FucAl was incorporated into pectic rhamnogalacturonan-I (RG-I). This labeled RG-I first localized to punctae in cell walls (Figure 2A), likely associated with locations where newly synthesized polysaccharides had been delivered via vesicles. More homogenous distribution of fluorescent label at later time points was posited to be the result of ongoing delivery of FucAl-labeled glycans to all parts of the cell wall or due to redistribution of glycans in the cell wall after delivery.

Treatment of plant tissues with clickable monolignol analogs has also been used for mapping cell wall lignification. Incorporation of both azido and alkynyl monolignol analogs in a single sample can be visualized using coupling of the azide label with a strained cyclooctyne fluorophore via strain-promoted alkyne-azide cycloaddition (SPAAC) with subsequent copper-catalyzed alkyne-azide cycloaddition (CuAAC) to attach a fluorophore to the alkyne label. Simon and coworkers used this technique to achieve labeling of flax stems and roots, noting that the fluorescent labeling was most intense in the outermost layers of the secondary xylem actively undergoing lignification (Figure 2H) [23]. Analysis of tissues in finer detail revealed that the different monolignol analogs were incorporated into specific cell wall domains of the same cells to different extents. Future development of these and other monolignol metabolic labeling probes may be useful for selectively labeling lignin and/or selected cell wall regions. Beyond plant development and growth, these approaches may also be able to provide interesting new insights into the impacts of rhizosphere bacteria on root tissues and microbial interactions with plant root cells at the subcellular level [33].

Metabolic labeling of proteins using bioorthogonal noncanonical amino acid tagging (BONCAT) has been applied in both plant tissues and soil microbial communities to investigate protein synthesis under various conditions and identify metabolically active community members from complex microbiome samples [27,28]. Azidohomoalanine (AHA) and homopropargylglycine (HPG), the most commonly used reagents for BONCAT, are readily available from commercial vendors. AHA and HPG both act as methionine analogs and are incorporated into proteins during translation under nutrient limitation. Importantly, introduction of noncanonical amino acids into newly synthesized proteins may alter protein functions as well as metabolic state, although use of these reagents at relatively low concentrations and for short durations (1–2 cell divisions) minimizes the impact [29]. To date, there are a very few examples demonstrating the use of BONCAT for specifically probing plant–microbe interactions, although BONCAT has been applied to elucidating plant stress responses toward light stress, osmotic shock, salt stress, and heat stress [27]. This work highlights the potential utility of BONCAT-based approaches for examining how plant-associated microbes influence translation in the plant host. Notably, studies of microbiome-host interactions using BONCAT may require adapted labeling protocols depending on the nutrient status of the system.

In addition to amino acids, monolignol compounds, and sugars, researchers have also applied click chemistry fatty acid tools for metabolic labeling in plants. Incorporation of propargylcholine into choline phospholipids was demonstrated for root, leaf, stem, silique, and seed tissues in *Arabidopsis thaliana* by Paper et al. [34]. Propargylcholine was taken up readily by roots, and click chemistry attachment of fluorophores enabled the authors to visualize the subcellular localization of these labeled phospholipids. In seedlings and mature plants, roots were labeled strongly, with propargylcholine accounting for 50% of the total choline and propargylcholine lipids, as determined by mass spectrometry. Incorporation in other tissues ranged from ~13–28%. For imaging applications, the researchers noted that labeling with clickable fluorophores such as fluorescein azide was limited to tissue surface layers; peeling or sectioning of tissues was required to visualize propargylcholine incorporation into internal tissue regions.

In a different study, Boyle and coworkers used ω-alkynyl fatty acids—Alk12, Alk14, and Alk16 as analogs of myristic, palmitic, and stearic acids, respectively—to investigate lipid modifications to specific proteins expressed in different plant model systems (Figure 3) [30]. Fatty acylation of proteins is involved in plant immune responses as well as pathogen virulence. The fatty acid analogs were well tolerated and penetrated into whole leaf tissues when applied by syringe infiltration. S-acylation of key cysteine residues in the flagellin-sensitive pattern recognition receptor FLS2 was readily observed in *Arabidopsis* protoplasts. Tomato resistance protein Pto, expressed in *Nicotiana benthamiana* leaves, was labeled with Alk12, Alk14, or Alk16, and these reagents did not interfere with the ability of Pto to trigger programmed cell death in host cells in response to the *Pseudomonas syringae* pv. *tomato* effector AvrPto. Myristoylation of the glycine-2 (G2) residue in AvrPto by the host was also detected in a transgenic *Arabidopsis* line treated with Alk12, demonstrating the utility of this approach for probing lipid modifications to both plant and microbial proteins.

### 2.2. Activity-Based Probes for Functionally Profiling Plant-Microbe Systems

The growing application and potential of activity-based probes (ABPs) in plant sciences has been well reviewed by Morimoto and van der Hoorn [35]. Since ABPs are designed to react with specific classes of enzymes based on their mechanisms of action, prior knowledge of target enzyme structure or sequence and even the native substrate is not required. Multiple enzymes from a single complex sample can, therefore, be identified based on their function (Figure 4A). Thus, activity-based protein profiling (ABPP) can enable functional discovery in complex, poorly annotated biological systems, including plants and soil microbial communities [1,36,37]. Even in one of the most studied model plants, *Arabidopsis thaliana*, only ~77% of the protein-coding sequences have been assigned structured annotation [38], while functional annotation of soil microbial communities remains a major experimental and computational challenge [39,40,41]. In complement to metagenomic and metatranscriptomic approaches, ABPs offer important opportunities for functionally characterizing plants and their associated microbiomes.

ABPs have been developed for diverse enzymatic activities and protein targets [42]. ABPs for protease, glycosidase, and other hydrolase activities are among the most popular. The commercial availability of some of these reagents, as well as the generosity of chemical biology researchers in sharing noncommercial probes, have greatly expanded the use of ABPs in plant and microbial sciences. Structures of selected ABPs that have been used for probing plant-microbe interactions are shown in Figure 4B. At a minimum, ABP designs feature a reactive moiety that enables covalent labeling of the target and a reporter tag, which may be a fluorophore for fluorescence labeling, biotin for streptavidin affinity enrichment, or a click handle such as an azide or alkyne for downstream attachment of a reporter. To enhance selectivity of an ABP for a desired target class, the probe may include specific recognition motifs. For example, the probe ASM101 was designed to target vacuolar processing enzymes (VPEs), a subclass of cysteine proteases; VPE selectivity is enhanced by inclusion of an aza-asparagine at the P1 position, while cross-reactivity with papain-like cysteine proteases is reduced by incorporating a proline residue at the P2 position (Figure 4B) [43].

Effective application of ABPP to plant-microbe systems can elucidate the key enzymes and signaling pathways driving host-pathogen interactions and potentially assist in the identification of genes and proteins that may confer disease resistance or susceptibility. Buscaill et al. applied ABPs for glycosidases to investigate how pathogenic bacteria suppress the plant immune response [44]. The probe JJB70 was used to investigate changes in the activity of a putative β-galactosidase, BGAL1, in *Nicotiana benthamiana* infected with bacterial pathogen *Pseudomonas syringae*. ABPP of apoplastic fluids showed a ~19-fold decrease in JJB70 labeling of BGAL1 in infected plants compared to controls (Figure 5). Further investigation revealed the presence of a heat-stable, basic, <3 kDa inhibitor molecule produced by the bacterial pathogen that inhibits BGAL1, but not other host hydrolases. Taken together, these findings were used to propose a model for BGAL1′s role in plant host immunity and elucidate potential pathogen strategies for subverting host defense responses.

Proteases are also highly implicated in plant immune responses and defense against pathogens. A recent study by Paulus et al. used two different ABPs to examine the activation of Rcr3, a secreted papain-like cysteine protease (PLCP) involved in plant immune response, by the subtilisin class of serine proteases [45]. DCG-04, an ABP for cysteine proteases and PLCPs, was used on substitution mutants of Rcr3 to determine which active site cysteine residues were essential for catalytic activity; C154 was identified as a key residue, while C153 was not required for protease activity. Like other PLCPs, generation of the mature, active protease mRcr3 was thought to proceed by self-activation, involving unfolding of the protein under acidic conditions followed by cleavage of the prodomain from the protease domain. However, Paulus and colleagues observed that proteolytically inactive mutants still formed analogous mRcr3 isoforms after incubation with tomato apoplastic fluid (AF), suggesting that other proteases in AF can convert proRcr3 to mRcr3. To identify serine proteases that could be involved in proRcr3 processing, the authors used the fluorophosphonate probe FP-TAMRA for ABPP of tomato AF. Of the identified proteins, the subtilase (SBT) class of endopeptidases was confirmed to cleave proRcr3. Treatment with the SBT inhibitor EPI1, derived from a potato blight pathogen, inhibited the tomato SBT P69B and successfully blocked mRcr3 formation from proRcr3. Based on these findings, Paulus et al. suggested that inhibition of host SBTs by pathogens may indirectly prevent activation of Rcr3(-like) immune proteases, thereby contributing to increased virulence. This study exemplifies how different ABPs can be used for functionally profiling complex biological mixtures to identify key proteins involved in specific signaling pathways, providing a deeper mechanistic understanding of these processes.

A number of other studies have demonstrated the utility of protease ABPs in diverse systems, highlighting the broad applicability of these chemical tools for investigating plant–microbe interactions. A recent study by Franco et al. demonstrated the use of the FP-TAMRA probe for visualizing changes in serine protease activities in citrus infected with the bacterium *Candidatus Liberibacter asiaticus* (CLas)*,* which causes Huanglongbing, also known as citrus greening disease [46]. Previous work using the DCG-04 probe to examine this citrus disease demonstrated that the CLas secreted protein Sec-delivered effector 1 (SDE1) inhibited host PLCP CsRD21a and other PLCP activities, suggesting this pathogen effector has a key role in infection [47]. Another PLCP ABP, MV201, has been used for probing plant response toward the 4E02 effector of the sugar beet cyst nematode *Heterodera schachtii* in *Arabidopsis* [48]. Vacuolar processing enzymes (VPEs) [43] and serine hydrolases were also studied in this pathogen-plant system using the aza-epoxide probe AMS101 and a fluorophosphonate with rhodamine reporter tag, respectively [49]. Similarly, DCG-04 and FP-biotin were used to examine protease activities in disease-resistant and -susceptible varieties of tomato infected with the soil-borne bacterial pathogen *Ralstonia solanacearum* [50]. ABPs have also been applied to profiling fungal enzymes [51] and enzyme activities in response to fungal species, including pathogens such as *Cladosporium fulvum* [52,53]. Although the above examples feature ABPP in probing the mechanisms of infection and disease progression in plant-pathogen interactions, ABPs will undoubtedly also be valuable for understanding beneficial plant–microbe associations that can be used to promote plant resilience under environmental stress or disease.

## 3. Challenges for Chemical Biology in Plant-Microbe Interactions

Although chemical biology is making significant strides forward into systems of increasing complexity, challenges remain ahead for researchers applying these chemical tools to plant-microbe systems.

### 3.1. Applying Chemical Biology Tools in Laboratory Models vs. Field Systems

One of the greatest challenges in the future study of plant–microbe interactions is the increasing need to connect laboratory studies to the field. Many of the studies highlighted in this review were performed on aerial plant tissues or in plants grown in artificial substrates. Unsurprisingly, there are far fewer examples of chemical biology investigations of plant roots in natural soils, despite the abundance of biological activity occurring in the rhizosphere. Soil is incredibly complex in both its physical and chemical properties; it is notoriously difficult to sterilize [54], and it is well recognized that diverse synthetic chemicals, such as pesticides, adsorb to soil and may undergo both biotic and abiotic transformation [55,56] making the controlled introduction of synthetic chemical probe compounds to a plant-microbe-soil system especially problematic.

Models for plant–microbe interaction studies often rely on artificial growth substrates, including hydroponic, semihydroponic, and gel-based growth media that can be sterilized, making axenic plant and microbe cultures possible. Separation of delicate plant roots from these substrates during sampling is often easier than isolating roots that have been grown in natural soils. However, artificial substrates often dramatically alter root morphology in plants [57]; although easy to use, a major shortcoming of these substitutes is their relative homogeneity, while heterogeneity in soils is crucial to the unique microhabitats and complex microbial relationships observed in soils and the plant rhizosphere [58,59]. Nevertheless, given the challenges of working with soil, plants grown on artificial substrates remain essential to plant-microbe studies. Further exploration of soilless media for plant growth, including refinement of artificial soil formulations that more closely mimic the structure and geochemistry of soil [60], are needed to develop more realistic plant-microbe-soil models and help to bridge the gap between laboratory and field studies.

Continued progress in soil microbiome research will also contribute to improved models for studying plants and their associated microbiomes, including expansion of single strain models to microbial consortia to communities [8]. Ultimately, while laboratory experiments will provide essential data for understanding the molecular mechanisms underpinning diverse plant–microbe interactions, there remains a very keen interest in analyzing field samples that reflect true environmental conditions. Chemical biology approaches have not yet been extensively applied to field studies of plant-microbe interactions; further optimization of these chemical tools and methods will make broader field deployment possible in the future.

### 3.2. Delivering Chemical Tools to Live Plant Tissues

Uniform application of chemical tools to plant tissues remains a significant complication. Given the intricate spatial organization of specific cellular types in plant tissues, in vivo labeling would be ideal, but the robust barriers presented by plant cells and tissues makes it difficult to predict passive or active uptake of a probe compound. Syringe infiltration or simple bath application can be used for introducing probes to live, intact plant tissues, although achieving equal exposure of all cells to the probe during in vivo labeling is challenging [35]. When applicable, homogenization of plant tissues prior to probe labeling may produce more consistent results. However, any impact that subcellular chemical environment and spatial organization of proteins or signaling molecules may have on functional activity is lost during bulk tissue homogenization.

Delivery of chemical probes in aqueous solution may also pose a challenge for investigating drought-related changes in plants. Buffers containing polyethylene glycol (PEG) or mannitol have been used previously to trigger osmotic stress in plants to simulate drought stress; this strategy was used by Glenn and colleagues to examine changes in protein synthesis via BONCAT [27].

Exploration of new delivery methods for probes targeted to above and below ground tissues may provide additional opportunities for probing intact tissues. Fichman, Miller, and Mittler have developed a fumigation approach wherein fluorescent indicators for reactive oxygen species were nebulized and taken up by plant leaves and stems [61]. This technique may provide an alternative to bath application of probes, although a period of high humidity is needed to promote open stomata for aerosolized probe uptake and therefore likely precludes application to drought-specific studies.

## 4. Conclusions

The technologies and approaches summarized in this review are only a small selection of the chemical biology tools available for probing complex interkingdom interactions. In conjunction with genetic and biochemical analyses, these tools will continue to improve our understanding of plant-microbe systems. While much of the work in this review has focused on pathogenic bacterial and fungal interactions with plants, future efforts to understand plant growth-promoting microbes may also benefit from the use of these and other chemical biology tools.

Additional development of chemical probes for key signaling molecules and phytohormones is still needed to better understand plant–microbe interactions, particularly beneficial interkingdom relationships. Already, applications of clickable N-acyl-homoserine lactone (AHL) analogs [62], auxin analogs [63], and other [64] fluorescent indicators are demonstrating the many potential ways that chemical tools can be used to probe plant-microbe systems [12,65]. Chemical inhibitors of specific pathways involved in development, defense, stress responses, and more are being developed, including molecules for probing the jasmonate and abscisic acid (ABA) signaling pathways [66,67]. Continued production and application of these tools will yield exciting new data elucidating these intricate interkingdom interactions. Further development, application, and optimization of chemical biology approaches to probing plant-microbe systems will continue to contribute to our fundamental understanding of plants, bacteria, and fungi in diverse terrestrial ecosystems.

## Figures and Tables

**Figure 1 molecules-26-00243-f001:**
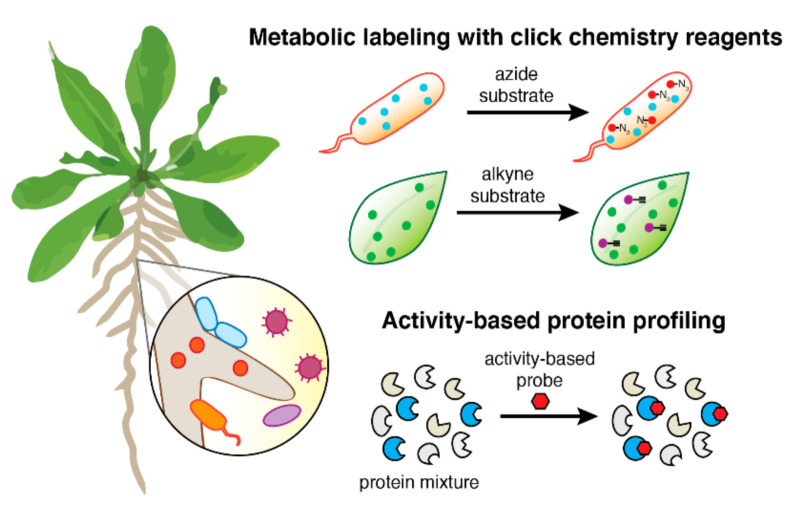
Click chemistry reagents and activity-based probes as chemical tools for studying plant-microbe interactions. Metabolic labeling of live microbes and plant tissues with azide- or alkyne-modified glycan, monolignol, lipid, or amino acid substrates enables click chemistry attachment of various tags for imaging or affinity enrichment. Activity-based probes label enzymes based on their specific mechanisms of action, allowing for functional characterization of proteins even in complex, poorly annotated biological samples.

**Figure 2 molecules-26-00243-f002:**
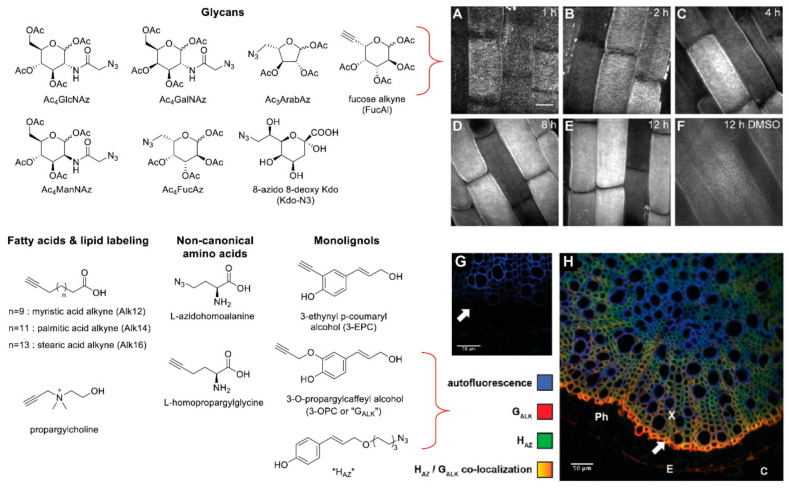
Structures and applications of selected azido and alkynyl reagents for metabolic labeling of plants and microbes. (**A**–**E**) Four-day-old *Arabidopsis* seedlings were treated with 2.5 μM FucAl for the indicated times, labeled with Alexa 488-azide, and z series of elongation-zone root epidermal cells were recorded using a spinning disk confocal microscope with a 1.4 NA 100× oil-immersion objective. Images are contrast enhanced maximum projections of the z series. (**F**) Control seedlings treated with DMSO for 12 h and labeled with Alexa 488-azide show background fluorescence. (Scale bars, 10 μm) [25]. (**G**,**H**) Bioorthogonal Labeling Imaging Sequential Strategy (BLISS) illustrating incorporation of monolignol chemical reporters “G_ALK_” and “H_AZ_” into cell walls in 2-month-old flax roots. Sectioned flax roots previously incubated with native p-coumaryl and coniferyl alcohols as negative control (G) or with azide-labeled p-coumaryl alcohol (H_AZ_) and alkyne-labeled coniferyl alcohol (G_ALK_) monolignol reporters (H) and observed by confocal microscopy [23]. Images and portions of the caption reprinted (adapted) from Anderson, C.T.; Wallace, I.S.; Somerville; C.R*. Proc. Natl. Acad. Sci. USA*
**2012**, *109*, 1329–1334 under the PNAS License to Publish, and from Simon, C.; Lion, C.; Huss, B.; Blervacq, A.S.; Spriet, C.; Guérardel, Y.; Biot, C.; Hawkins, S. *Plant Signal. Behav*. **2017**, *12*, e1359366 under the terms of the Creative Commons Attribution-NonCommercial-NoDerivatives License (http://creativecommons.org/licenses/by-nc-nd/4.0/).

**Figure 3 molecules-26-00243-f003:**
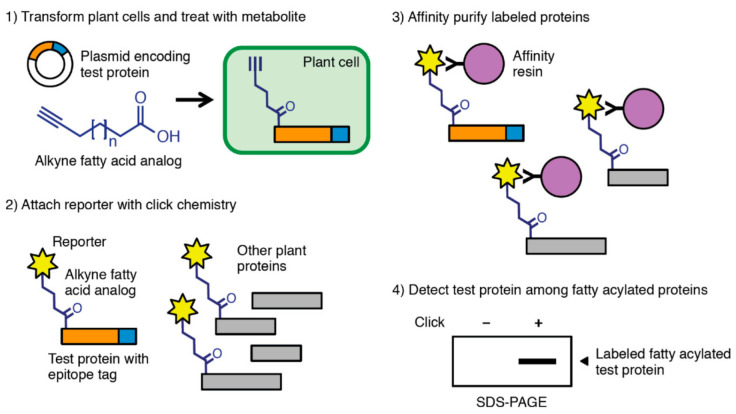
Schematic of metabolic labeling using alkyne fatty acid analogs for the capture and enrichment of fatty acylated proteins for investigating molecular mechanisms of plant immunity and pathogen virulence. Boyle et al. demonstrated this approach in *Arabidopsis* protoplasts and stable transgenic *Arabidopsis* plants using well-studied proteins FLS2, AvrPto, and Pto, which are involved in plant–pathogen interactions [30]. Image and caption reprinted (adapted) from Boyle, P.C.; Schwizer, S.; Hind, S.R.; Kraus, C.M.; De la Torre Diaz, S.; He, B.; Martin, G.B. *Plant Methods*
**2016**, *12*, 38 (DOI: 10.1186/s13007-016-0138-2) under the terms of the Creative Commons Attribution 4.0 International License (http://creativecommons.org/licenses/by/4.0/).

**Figure 4 molecules-26-00243-f004:**
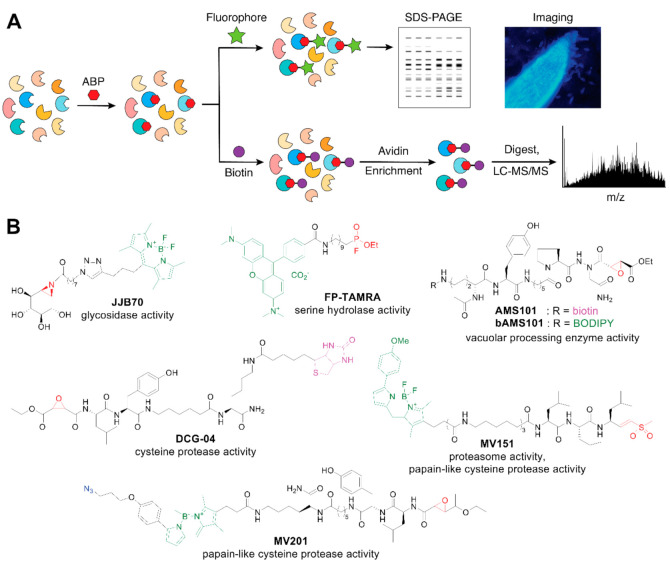
(**A**) Workflow for activity-based protein profiling using activity-based probes (ABPs; red hexagon) for covalent labeling of target proteins. Proteins labeled with fluorophore-conjugated ABPs can be analyzed by fluorescence gel electrophoresis or fluorescence microscopy. Enrichment of proteins labeled with biotin-conjugated ABPs, followed by trypsin digestion and liquid chromatography tandem mass spectrometry (LC-MS/MS) proteomics analysis, enables target identification. (**B**) Selected activity-based probes that have been applied to studying plant-microbe interactions. Functional activities targeted by each probe are shown below the probe names. Reactive groups are shown in red, click chemistry tags are shown in blue, fluorophores are shown in green, and affinity enrichment tags are shown in pink.

**Figure 5 molecules-26-00243-f005:**
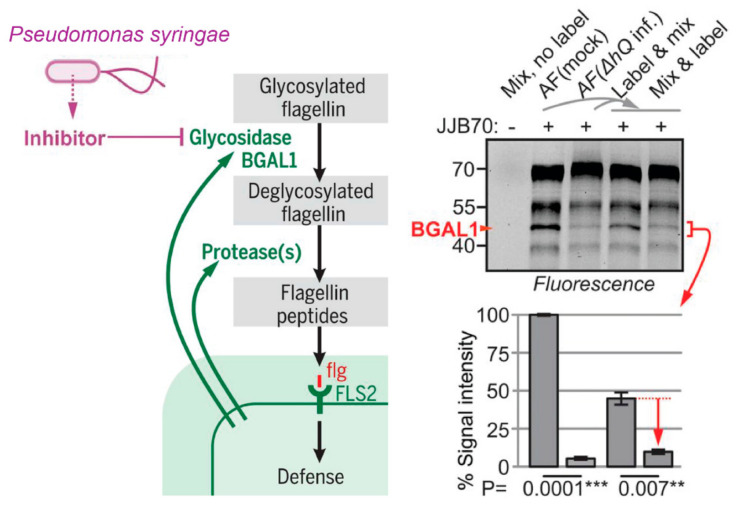
Secreted β-galactosidase BGAL1 and proteases contribute to plant host immunity against bacteria with glycosylated flagellin. Secretion of a BGAL1 inhibitor by pathogenic *P. syringae* suppresses host defense response. BGAL1 labeling is suppressed in apoplastic fluid (AF) of infected plants at 2 days postinfection (dpi). The two samples were also mixed in a 1:1 ratio before (mix and label) and after (label and mix) labeling with JJB70. The quantified fluorescence of the BGAL1 signal is plotted below. Error bars indicate mean ± SE of n = 3 replicates; *t* test was used to determine P values [44]; *** and ** indicate *p* values < 0.001 and 0.01, respectively. Image and caption adapted from Buscaill, P.; Chandrasekar, B.; Sanguankiattichai, N.; Kourelis, J.; Kaschani, F.; Thomas, E.L.; Morimoto, K.; Kaiser, M.; Preston, G.M.; Ichinose, Y.; van der Hoorn, R.A.L. *Science*
**2019**, *364*, 6436 (DOI: 10.1126/science.aav0748). Copyright 2019 American Association for the Advancement of Science.

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
