# Peer review of "Interrogating Plant-Microbe Interactions with Chemical Tools: Click Chemistry Reagents for Metabolic Labeling and Activity-Based Probes"

_molecules, 2021, doi:10.3390/molecules26010243_

Round 1

Reviewer 1 Report

The study of plant-microbial interactions is important from a fundamental and applied point of view. Plant and microbial objects are very informative for the study of evolution and co-evolution among the largest kingdoms in the Earth's biota and form the basis for a variety of industrial, food, environmental and biomedical biotechnologies. Therefore, the peer-reviewed review devoted to the latest methods of their research using biosensors based on click chemistry and bioorthogonal reactions is of undoubted interest for a wide range of interdisciplinary researchers, including the readers of the Molecules journal.

Minor remark:

The author should clarify the term "biosensor" used by him. Traditionally, a biosensor is a technical device that generates a signal in response to a biospecific (chemical) interaction it detects (Turner et al. Biosensors: fundamentals and applications. Oxford, UK: Oxford University Press, 1987, 770 p .; Dincer et al. Adv. Mater. 2019. Vol. 31, No. 30 (e1806739). P. 1-28. DOI: 10.1002/adma.201806739). In contrast, the author often refers to the biological (chemical) molecular components of such a device as a biosensor. For example, in the caption to Figure 1, the author writes that “Biosensors can be expressed in plant tissues as well as in bacterial cells ...”. However, in my opinion, it would be more accurate to say that "Key components of biosensors can be expressed in plant tissues as well as in bacterial cells ...".

Author Response

We thank the reviewer for his/her time in reviewing this manuscript.

Based on commentary from another reviewer, I have decided to remove the section on biosensors, given the large number of existing reviews solely on biosensors in plant and microbe systems--I believe the biosensor topic would be best served if fully reviewed in a dedicated manuscript, rather than only partially and cursorily reviewed in my original submission. I am grateful for the reviewer's recommendation for clarifying how biosensor is defined, and will be certain to keep this in mind in any future writing endeavors on the topic.

Given the removal of the biosensor section, I have spent some additional time on the click chemistry reagents and activity-based probe sections, which I hope provides the review with a stronger overall theme on small molecule chemical tools for studying plant-microbe interactions.

Reviewer 2 Report

General comments

In the manuscript, Lin VS reviews the chemical biology tools to study plant-microbe interactions. In particular biosensors and click chemistry reagents are presented and reviewed.

Since the review is mostly focused on those elements, the title of the review is not appropriate and it should be more precise for interested researchers.

Besides, often, along with the manuscript the authors avoid the presentation of important aspects and applications of certain chemical tools just referring to other preview reviews that are so missing in the present manuscript. The author should avoid such a kind of simplification and briefly present the data. On the other hand, some examples are too exhaustive (see the study by Del Valle et al.,).

More complete and better discussed is the section on the bioorthogonal click chemistry reagents.

Specific comments concern:

Line 13 Eliminate « have »

Line 21 Replace « single-celled » with « single-cell »

Legend Figure 1 the references should be included in the references section. The only short abbreviation should be cited in the legend.  

Line 139 The sentence is unfinished

Line 156 Eliminate the double points

All along with the manuscript, the references should be located before the punctuation.

And all the text should be justified.

Author Response

I would like to thank the reviewer very much for his/her time in reviewing this manuscript.

The reviewer is absolutely correct in identifying significant weaknesses in the biosensor section; given the large number of excellent research papers and existing reviews on biosensors, this topic was not covered sufficiently in my original submission.

In consideration for the manuscript length, I have removed the biosensor section so that it can be more appropriately and thoroughly reviewed on another occasion. I have adjusted the scope of this review to focus solely on the metabolic labeling and activity-based protein profiling. Additional details are now provided in these sections to address the reviewer's concerns about oversimplification of the presented data. The title has been accordingly corrected and made more precise, also based on the reviewer's recommendation. I hope these changes provide the review with a stronger overall theme on small molecule chemical tools for studying plant-microbe interactions and improve the quality of the manuscript.

I again thank the reviewer for his/her thorough editing of this manuscript and attention to detail. The following typographical errors have been corrected:

Line 13 Eliminate « have » Done.

Line 21 Replace « single-celled » with « single-cell » Done.

Legend Figure 1 the references should be included in the references section. The only short abbreviation should be cited in the legend. Figure has been removed. 

Line 139 The sentence is unfinished. This sentence has been removed.

Line 156 Eliminate the double points. Done.

All along with the manuscript, the references should be located before the punctuation. Done.

And all the text should be justified. Done.

Round 2

Reviewer 2 Report

General comments

The authors modified the manuscript making it more readable and concise.

The reviewed data could be of interest for the scientific community interested on click chemistry reagents. The revised paper is now suitable for publication in Molecules Journal.